# Transferrin-Decorated Niosomes with Integrated InP/ZnS Quantum Dots and Magnetic Iron Oxide Nanoparticles: Dual Targeting and Imaging of Glioma

**DOI:** 10.3390/ijms22094556

**Published:** 2021-04-27

**Authors:** Didem Ag Seleci, Viktor Maurer, Firat Baris Barlas, Julian Cedric Porsiel, Bilal Temel, Elcin Ceylan, Suna Timur, Frank Stahl, Thomas Scheper, Georg Garnweitner

**Affiliations:** 1Institute for Particle Technology (iPAT), Technische Universität Braunschweig, 38104 Braunschweig, Germany; d.ag-seleci@tu-braunschweig.de (D.A.S.); v.maurer@tu-braunschweig.de (V.M.); c.porsiel@tu-braunschweig.de (J.C.P.); b.temel@tu-braunschweig.de (B.T.); 2Center of Pharmaceutical Engineering (PVZ), Technische Universität Braunschweig, 38106 Braunschweig, Germany; 3Faculty of Science Biochemistry Department, Ege University, 35100 Izmir, Turkey; f.b.barlas@gmail.com (F.B.B.); elcinceylan5@gmail.com (E.C.); suna.timur@ege.edu.tr (S.T.); 4Institute for Technical Chemistry, Leibniz University Hannover, 30167 Hannover, Germany; stahl@iftc.uni-hannover.de (F.S.); scheper@iftc.uni-hannover.de (T.S.)

**Keywords:** multifunctional niosomes, quantum dots, iron oxide nanoparticles, glioma imaging

## Abstract

The development of multifunctional nanoscale systems that can mediate efficient tumor targeting, together with high cellular internalization, is crucial for the diagnosis of glioma. The combination of imaging agents into one platform provides dual imaging and allows further surface modification with targeting ligands for specific glioma detection. Herein, transferrin (Tf)-decorated niosomes with integrated magnetic iron oxide nanoparticles (MIONs) and quantum dots (QDs) were formulated (PEGNIO/QDs/MIONs/Tf) for efficient imaging of glioma, supported by magnetic and active targeting. Transmission electron microscopy confirmed the complete co-encapsulation of MIONs and QDs in the niosomes. Flow cytometry analysis demonstrated enhanced cellular uptake of the niosomal formulation by glioma cells. In vitro imaging studies showed that PEGNIO/QDs/MIONs/Tf produces an obvious negative-contrast enhancement effect on glioma cells by magnetic resonance imaging (MRI) and also improved fluorescence intensity under fluorescence microscopy. This novel platform represents the first niosome-based system which combines magnetic nanoparticles and QDs, and has application potential in dual-targeted imaging of glioma.

## 1. Introduction

Glioma are one of medicine’s most challenging problems and lead to a poor prognosis for the patient with a mean survival time of less than 18 months [1]. Only 3–10% of the patients survive for more than 5 years following diagnosis [2]. Surgical resection, chemotherapy, radiotherapy and novel therapeutic modalities such as immunotherapy, gene therapy, and photodynamic therapy are current treatments for glioma [3]. Surgery is the most effective treatment to increase the survival time of glioma patients. However, the operation bears high risk due to the localization of tumor tissue in the brain, and ineffective resection leads to remaining glioma cells that can proliferate and migrate quickly. An effective tumor resection needs novel imaging techniques for the detection of tumor localization and surgical phases. The commonly used imaging techniques are magnetic resonance imaging (MRI) and computed tomography (CT) [4,5]. Although images from CT and MRI can reveal the location of the tumor, it remains difficult to differentiate neoplastic and normal brain tissue because of the poor visual contrast.

Advances in nanotechnology have triggered a growing interest for the use of nanomaterials in medicine to solve a number of issues associated with therapy and diagnosis. Therapeutic agents have been combined with nanoparticles to minimize toxicity, increase their solubility, prolong the circulation half-life, reduce their immunogenicity, and improve their distribution [6,7,8]. Multifunctional nanomaterials such as quantum dots (QDs) and polymers have been investigated and have shown great promise in targeted imaging of cancer cells [9,10,11].

Magnetic iron oxide nanoparticles (MIONs) are extensively used as MRI contrast agents. Dependent on the specific relaxation process, the resulting magnetic resonance (MR) images are classified into longitudinal relaxation time weighted images (T1 contrast) and transverse relaxation time weighted images (T2 contrast). MIONs with a diameter below 4 nm are preferred as T1 (positive) contrast agents, while nanoparticles with a diameter greater than 4 nm are used as T2 (negative) contrast agents [12]. MRI-optical dual imaging is a promising approach that can provide more accurate diagnosis than any single imaging modality. In this approach, MRI provides noninvasive in vivo high-resolution anatomical images, while fluorescence imaging can deliver microscopic information in postmortem pathological tissues [13]. Fluorescent dyes such as rhodamine, fluorescein, and conjugated polymers have been used to design a dual-mode probe for biomedical applications [14,15,16]. Jang et al. presented the synthesis and the use of monodisperse iron oxide nanoparticles coated with fluorescent silica nano-shells for fluorescence and magnetic resonance dual imaging of tumors. The in vitro and in vivo data presented in that study demonstrate the high potential of the designed core-shell nanoparticles as biocompatible dual contrast agents for in vivo biomedical imaging [17]. Although successful cell labeling has been achieved with organic dyes, they are plagued by low quantum yields and photobleaching. Functionalized fluorescent nanoparticles, such as QDs, are promising alternatives for bioimaging [18], showing excellent physical and fluorescent properties, such as size-dependent broad absorption spectra and narrow emission spectra, high quantum yields, resistance to photobleaching, and high photochemical stability [19,20]. However, there is strong concern about the cytotoxicity of QDs, especially for the commonly used cadmium-based materials (in particular, CdSe, and CdS). Numerous studies have demonstrated that these cadmium-based QDs can cause significant DNA damage due to acute toxic effects [21]. Although the cytotoxic effects of cadmium-based QDs can be reduced by controlling the size and the surface ligand chemistry of the particles [22], the use of alternative QDs is preferred for biomedical applications. Indium phosphide (InP) QDs show similar size-dependent optical properties, resistance to photobleaching, and tunable fluorescence to their Cd-based counterparts, but they are less toxic [23]. The degree of toxicity of QDs also depends on their colloidal stability; therefore, the deposition of a ZnS shell was postulated to provide advantages to avoid, or at least slow down, the decomposition of QDs in intracellular environments [24]. Lin et al. have studied the in vivo long-term toxicity of InP/ZnS QDs in BALB/c mice, and observed no acute toxic effects according to hematology, blood biochemistry, and histological analysis [25]. Hence, core/shell-structured InP/ZnS QDs have become popular alternatives to cadmium-containing nanomaterials for biomedical applications. However, QDs are mostly produced in organic solvents, making them unsuitable for direct use in biomedical and clinical applications. To overcome this limitation, such QDs can be inserted into lipid bilayers to enhance their hydrophilicity, stability, and biocompatibility [26,27,28].

Integration of different diagnostic agents into one platform, the so-called “all-in-one” approach, has been postulated as a diagnostic tool for glioma with high potential [29]. Thereby, multifunctional probes can be created by combining MRI agents, fluorescence imaging agents, and photoacoustic imaging agents by using a suitable carrier system [30]. Niosomes are vesicular carriers which provide a bilayer structure that is promising for the design of all-in-one diagnostic systems. Various agents with a wide range of solubilities could be entrapped in the aqueous core or between the membrane bilayers of these structures [8,31]. Furthermore, this approach allows us to combine crucial properties into one system, such as tumor active targeting, imaging, and specific therapy, which more effectively supplies multimodal methods to combat glioma [32]. Active targeting is promising and can be achieved with the specific binding of targeting ligands conjugated on nanoparticles to receptors overexpressed on the cancer cells [33].

Among others, the transferrin (Tf) receptor is one of the most effective and most commonly used target receptors on cellular surfaces [34]. Tf is an iron-binding blood plasma glycoprotein that controls the level of free iron in biological fluids, while also facilitating the transport of iron to proliferating cells. Tf is internalized into the cells via Tf-receptor-mediated endocytosis. The Tf-receptor is highly expressed in glioma cells because of their rapid proliferation rate as well as large iron demand, whereby tumor xenografts which were generated from cancer cell lines retain the Tf-receptor expression in vivo [35,36]. Therefore, the ability of Tf to internalize via the Tf-receptor, as well as the high level of expression on cancer cells, make it an attractive target for selective delivery of diagnostic and therapeutic agents, especially to brain tumors. Several types of nanoparticles were functionalized with Tf-receptor-binding ligands such as peptides, antibodies, or Tf itself, to deliver therapeutics to the brain [37,38]. Furthermore, active targeting can be combined with physical targeting, such as the application of a magnetic field, light, or ultrasound, for efficient glioma targeting via the use of suitable functional nanoparticles.

Here, magnetic iron oxide nanoparticles (MIONs) and InP/ZnS QDs were synthesized and integrated into niosomes. Subsequently, Tf was conjugated to the surface of the niosomes for dual targeting and imaging of glioma. The designed PEGNIO/QDs/MIONs/Tf represents the first niosome-based system combining magnetic nanoparticles and QDs. The physicochemical parameters of the individual components as well as the assembled niosomes, in particular their size, surface charge, and morphology, were investigated in detail. Bioinvestigations, such as cytotoxicity and cellular uptake of the dual imaging systems, were examined with the U87 glioblastoma cell line. The glioma-targeted imaging capability of the nanoparticle-encapsulated niosomes was carefully evaluated under an external magnetic field. The dual imaging of glioma, including MR imaging and fluorescence imaging, was also confirmed in vitro.

## 2. Results and Discussion

### 2.1. Characterization of CA-MIONs and InP/ZnS QDs

The synthesis of MIONs was performed by co-precipitation of Fe^2+^ and Fe^3+^ in alkaline conditions. TGA was performed on dried samples before and after citrate coating (Figure 1A). While the net weight loss for the neat MIONs is about 7.0%, the weight loss for CA-MIONs amounts to 11.7%, which proves the presence of CA on the MIONs and thus successful modification [39]. Moreover, a TEM image of CA-MIONs is shown in Figure 1B, revealing a spherical morphology and relatively uniform size of the obtained particles. XRD was carried out to determine the phase content and crystallinity of the obtained products (Appendix A). The sample presents distinct Bragg reflexes that can be correlated with the reference patterns of magnetite (ICSD: 98-015-8714) and maghemite (ICSD: 98-007-9196). However, it is not possible to distinguish between maghemite and magnetite, as both materials have the same spinel inverse crystal structure; hence, determination of the composition is not possible by XRD analysis [40]. Using the Debye-Scherrer formula, a crystallite size of about 10 nm was calculated for CA-MIONs which is consistent with the particle size obtained by TEM. Appendix A shows the FTIR spectra of CA, MIONs and CA-MIONs. The peak at 1710 cm^−1^ is attributed to the C=O vibration in CA. This band shifts to 1600 cm^−1^ in CA-MIONs due to the binding of CA to the magnetite surface. Carboxylate groups of CA should complex with the Fe atoms on the magnetite surface and render a partial single bond character to the C=O bond, thus weakening it and shifting the stretching vibration frequency to a lower value [41]. The superparamagnetic character of the CA-MIONs is demonstrated by the magnetization curve (Appendix A), revealing a saturation magnetization of 37.1 Am^2^/kg. In a recent study, we showed that the encapsulation of MIONs into the niosomal structure was achieved with an entrapment efficiency of approx. 30%, which results in the formation of superparamagnetic niosomes with reduced saturation magnetization, but also an efficient magnetic targeting capability [42]. Furthermore, the zeta potential and hydrodynamic size of CA-MIONs were measured as −20.3 mV and 18.7 nm (Appendix A), respectively. 

The optical properties of the obtained InP/ZnS QDs were characterized by UV/Vis and fluorescence spectrophotometry. The fluorescence and extinction spectra of the QDs are shown in Figure 1C. The QDs show orange emission (λ_em_ = 600 nm) with an excitation wavelength of 488 nm. The TEM image of InP/ZnS QDs (Figure 1D) demonstrates a relatively monodisperse size distribution with an average size of ∼2.5 nm and mostly trigonal shape.

### 2.2. Characterization of Niosomes with Integrated InP/ZnS QDs and CA-MIONs 

The physicochemical properties of vesicular carrier systems, in particular their size, shape, and surface chemistry, play a critical role in determining tissue penetration and cellular delivery [9]. The particle size, electrical potential at the shear plane, and morphology of the prepared samples were characterized by DLS, zeta potential measurements, and TEM images. PEGNIO/QDs/MIONs have a negative zeta potential of −29.3 mV. Coupling of the positively charged Tf protein increased the surface potential of the niosomes, leading to a zeta potential of −19.8 mV [37]. The average particle size of PEGNIO/QDs/MIONs/Tf was detected to be 178.5 nm by DLS, which furthermore revealed a monomodal particle size distribution with a PDI of 0.192, demonstrating that the vesicle population is relatively homogeneous in size (Appendix A). Table 1 shows an overview for the hydrodynamic size, PDI and zeta potential of CA-MIONs and the different PEGNIO formulations [6,7]. The morphology of PEGNIO/QDs/MIONs was observed via TEM images at 120 keV as spherical (Figure 2A). To detect the encapsulated QDs and MIONs, the measuring voltage was increased to 200 keV, which destroyed the vesicle membrane. The particles inside the vesicles show identical morphology (Figure 2B) to that observed by TEM images of free QDs or free MIONs (Figure 1B,D). Additionally, to confirm the presence of both InP/ZnS QDs and MIONs in the niosomes, a photograph of PEGNIO/QDs/MIONs was taken under illumination by UV light and the migration of PEGNIO/QDs/MIONs, when applying a permanent magnet with a magnetic field of 1.3 T, was investigated (Appendix A). Therefore, it is clear that the PEGNIO/QDs/MIONs exhibit fluorescence signals and migrate in the direction of the magnetic field. These results prove that QDs and MIONs were encapsulated into the PEGNIO. The stability of PEGNIO/QDs/MIONs/Tf was tested via DLS analysis and no changes were observed in the size and PDI values after one month of storage at 4 °C in the dark (data not shown). 

Additionally, the sample was diluted in cell culture media and was incubated at 37 °C for 24 h. The size of the sample was measured before and after incubation, and no changes were observed. The photometric characteristics of InP/ZnS QD after encapsulation were examined by fluorescence spectrophotometry. As shown in Appendix A, the fluorescence emission spectrum of PEGNIO/QDs/MIONs was similar to that of InP/ZnS QDs. Encapsulation into niosomes reduced the fluorescence intensity of the QDs to some extent, possibly due minor absorption effects of the niosome shell or the MIONs, but it did not result in a shift of the emission wavelength.

### 2.3. Cytotoxicity 

A CTB assay was used to determine the cytotoxicity of PEGNIO/QDs/MIONs and PEGNIO/QDs/MIONs/Tf on U87 cells in the presence or absence of an external magnetic field. The cells were incubated with different concentrations of PEGNIO/QDs/MIONs and PEGNIO/QDs/MIONs/Tf for 24 h. The ratio of all niosome components was fixed, the given Fe concentration is a measure for the overall concentration of the niosome system. The obtained values of cell viability are presented in Figure 3. The cytotoxicity of PEGNIO/QDs/MIONs and PEGNIO/QDs/MIONs/Tf with magnet treatment (MT) was stronger than without MT. The calculated half inhibition concentration (IC_50_) of PEGNIO/QDs/MIONs was 0.50 mM and 0.31 mM (*p* < 0.05) for the systems without MT and with MT, respectively. The conjugation of the targeting ligand to the PEGNIO/QDs/MIONs increased their toxicity to U87 cells. IC_50_ values were calculated to be 0.39 mM (without MT) and 0.23 mM (with MT) for PEGNIO/QDs/MIONs/Tf (*p* < 0.05). This may be due to the dual targeting effect of PEGNIO/QDs/MIONs/Tf, which were mainly internalized by receptor-mediated endocytosis. The presence of external magnetic fields supports the attachment of niosomes alongside the cell layer, providing a higher availability of the nanoparticles at the target sites.

### 2.4. Cellular Internalization and Uptake 

Tf has previously been shown to be a promising targeting ligand with high specificity and affinity for glioma [37,43,44,45,46]. The cellular internalization of PEGNIO/QDs/MIONs and PEGNIO/QDs/MIONs/Tf was determined via fluorescence microscopy studies. The niosome dispersions were diluted with medium and then added to the cells. To evaluate the effect of an external magnetic field on cellular uptake, a 1.3 T permanent magnet was placed under the cell-growth plate for 15 min. Afterward, the cells were incubated for 4 h at 37 °C without an external magnetic field and washed once in PBS. Fluorescence microscopy revealed that QDs were localized in both the cytosol and nuclear envelope of glioma cells (Figure 4), evidencing the successful delivery of QDs via the niosomal system. In a similar vein to our results, biocompatible cationic InP/ZnS QDs were applied to breast cells in an earlier study by Devatha et al. and confocal microscopy images showed the fluorescence of InP/ZnS QDs inside the cytosol of breast cancer cells [47].

Non-specific cellular internalization of PEGNIO/QDs/MIONs was observed in glioma cells (Figure 4A), however, cellular internalization showed a 2.4-fold enhancement in relative fluorescent units (RFU) due to the magnetic treatment (Figure 4B). Modifying the niosome surface with Tf allows for a 2.7-fold higher cellular uptake through Tf-receptor-mediated endocytosis (Figure 4C) compared to plain cellular internalization of PEGNIO/QDs/MIONs. In particular, the application of PEGNIO/QDs/MIONs/Tf together with an external magnetic field resulted in a 3.1-fold enhancement in RFU due to the cumulative effect of magnetic guidance and the enhanced Tf-receptor-based attachment on the cell surface (Figure 4D). These results clearly demonstrate the efficient dual targeting modality of the niosomal system.

Furthermore, flow cytometry was used to additionally evaluate the receptor-mediated cell targeting efficacy of niosomes on U87 cells. The cells were treated with PEGNIO/QDs/MIONs and PEGNIO/QDs/MIONs/Tf for 2 h. Untreated control cells and treated cells were analyzed using a BD Accuri C6 flow cytometer. Figure 5 indicates that PEGNIO/QDs/MIONs/Tf was taken up by U87 cells specifically. Tf is able to selectively bind to tumor cells via the Tf receptor, which is overexpressed in glioma cells [48]. The higher expression of Tf receptors on U87 cells enables a much greater proportion of targeted niosomes to enter the cells via receptor-mediated endocytosis, in comparison to non-targeted niosomes. We verified this Tf-receptor-mediated endocytosis of Tf-functionalized PEGNIO formulations by Tf-receptor-expressing U87 glioma cells via performing flow-cytometric cellular uptake measurements of fluorescence-labeled PEGNIO/Tf in Tf-receptor negative cells (A549, which expresses hardly any Tf-receptors [49]) (Appendix A). Because, in this case, the internalization of PEGNIO and PEGNIO/Tf was shown to be almost uniform and therefore independent of Tf-functionalization, the presented enhanced internalization of Tf-labeled PEGNIO/QDs/MIONs by U87 cells can be attributed to a Tf-based endocytosis.

### 2.5. MR Imaging

MIONs are among the most used diagnostic tools in cancer, with T2-weighted MRI as a common application [50,51]. In addition to the fluorescence imaging technique, MRI was applied to the U87 cells after administration of the newly developed diagnostic nanoparticle system. To compare the effectiveness of PEGNIO/QDs/MIONs/Tf, the control group (cells without treatment) was also investigated via MRI (Figure 6A). Figure 6B presents an MR image after the incorporation of PEGNIO/QDs/MIONs/Tf into the glioma cells with an obvious negative-contrast enhancement. The contrast density difference between control and sample is shown in Figure 6C. These results prove that PEGNIO/QDs/MIONs/Tf has great potential to be a contrast and fluorescent dual imaging agent for glioma diagnosis.

## 3. Materials and Methods

### 3.1. Materials

1,2-distearoyl-sn-glycero-3-phosphoethanolamine-*N*-[maleimide(polyethyleneglycol)-2000] (DSPE-PEG(2000) maleimide) was provided by Avanti (Alabaster, AL, USA). Sorbitan monostearate (Span60), cholesterol, iron(II) chloride tetrahydrate (FeCl_2_·4H_2_O), iron(III) chloride hexahydrate (FeCl_3_·6H_2_O), citric acid (CA), indium(III) chloride (InCl_3_), zinc(II) chloride ZnCl_2_, tris(dimethylamino)phosphine (TDMAP), dodecanethiol (DDT), 4-(2-hydroxyethyl)-1-piperazineethanesulfonic acid (HEPES), ethylenediaminetetraacetic acid (EDTA), zinc stearate, oleylamine (OLA), Dulbecco’s Modified Eagle Medium (DMEM), transferrin (Tf), and 2-iminothiolane hydrochloride (Traut’s reagent) were purchased from Sigma-Aldrich (Munich, Germany). 

### 3.2. Synthesis and Characterization of CA-MIONs

Citrate-coated magnetic iron oxide nanoparticles (CA-MIONs) were synthesized via a co-precipitation method [41,52]. 0.75 g of FeCl_3_·6H_2_O and 0.375 g of FeCl_2_·4H_2_O were dissolved in 50 mL of distilled water and mixed at 80 °C under nitrogen atmosphere in a 100 mL three-neck flask equipped with a reflux condenser. After 10 min, 10 mL of NH_4_OH (28–32% aq.) was added dropwise to the reaction mixture. 1 h after the beginning of the reaction, a previously prepared CA solution (1.0 g in 2 mL of distilled water) was injected into the reaction mixture. The reaction temperature was increased to 95 °C and the system stirred for one further hour. Subsequently, the resulting nanoparticle dispersion was left to cool to room temperature and washed twice with deionized water via decantation under magnetic fixation of the nanoparticles. Additionally, the dispersion was purified for 24 h using a 12–14 kDa dialysis bag to remove excess unbound CA. 

### 3.3. Synthesis and Characterization of InP/ZnS QDs

The synthesis of stabilized zinc sulfide-coated indium phosphide (InP/ZnS) core-shell quantum dots was adapted from Ellis et al. with minor adjustments (see SI Section for details) [53]. 

### 3.4. Preparation of Niosomes Loaded with InP/ZnS QDs and CA-MIONs 

The niosomal nanoparticles (PEGNIO/QDs/MIONs) were prepared by the thin-film hydration method [54]. Span 60, cholesterol, and DSPE-PEG(2000) maleimide were dissolved in chloroform with the molar ratio of 4.95:4.95:0.1. Then, 100 µL of the InP/ZnS quantum dots dispersed in chloroform were added to the niosomal precursor solution in a round-bottom flask. The solvent was evaporated with constant rotation under reduced pressure to form a thin lipid film. Afterwards, the thin film was hydrated with 1.0 mL of CA-MIONs aqueous solution at 60 °C for 60 min. A schematic representation of the niosome synthesis and nanoparticle encapsulation processes is shown in Scheme 1. The prepared multilamellar vesicles were processed to form small unilamellar vesicles by extrusion (extruder provided by Avanti Polar Lipids, USA) through 0.4 µm and 0.1 µm pore size polycarbonate filters (Cytiva Life Sciences, Dassel, Germany). Subsequently, the niosome dispersion was centrifuged at 11,000 rpm for 15 min and the sediment redispersed in water to remove the excess non-incorporated quantum dots and CA-MIONs.

### 3.5. Preparation of Transferrin-Conjugated Niosomes

Tf-targeted niosomal nanoparticles (PEGNIO/QDs/MIONs/Tf) were prepared via the attachment of Tf to the niosome surface. This was achieved using 5 nmol Tf and 25 nmol 2-iminothiolane (Traut’s reagent) in 500 μL HEPES EDTA buffer, at pH 8.5 (30 mM HEPES and 0.1 mM EDTA). The mixture was incubated for 2 h at room temperature while shaking to complete the thiolation of Tf. Thiolated Tf was then washed with HEPES pH 6.5 in a concentration column (30 kDa, Amicon^®^ by Millipore) and concentrated to a final volume of 200 µL. The loaded Tf was immediately added to the niosomes and incubated for 24 h at 4 °C resulting in the formation of a thioether linkage [55]. The thiol group of Tf was coupled with the maleimide group of the PEG chains on the niosomes.

### 3.6. Characterization of Niosomes

The size and zeta potential of the prepared nanoparticles were measured by DLS analysis using the Zetasizer Nano ZS and the Zetasizer Nano software (v7.12) from Malvern Panalytical (Kassel, Germany). The polydispersity index (PDI) was used as a measure of the width of the size distribution. The measurements were performed at 23 °C with a 173° backscattering setup. A previous dilution of the samples by a factor of 10^4^–10^5^ minimized occurring fluorescence. To obtain the hydrodynamic diameter, the modal value of the respective intensity distribution was taken. The zeta potentials were attained using a capillary zeta cuvette (DTS1070C, Malvern Panalytical Ltd.).

The morphology of the niosomes after integration of the nanoparticles was monitored via transmission electron microscopy (TEM). Briefly, one drop of the niosome dispersion was applied onto a carbon film on a 3.05 mm woven copper net with 300 mesh (Plano GmbH, Wetzlar, Germany). The remaining liquid was removed by blotting onto filter paper. Then, staining with 2% aqueous phosphotungstic acid was performed without removing the excess, but while allowing evaporation. The samples were observed using a Tecnai G^2^ F20 TMP (turbo-molecular pump) from Fei (Hillsboro, OR, USA), operating at an accelerating voltage of 120 keV and 200 keV in a bright-field image mode. 

For the powder X-ray diffraction (XRD) analysis, an Empyrean series 2 from Malvern Panalytical (Kassel, Germany) with Cu-K_α_ radiation (wavelength λ of 0.154 nm) was used. The MIONs were dried in an oven, put onto a Si sample holder and measured in a range of 20°–90° 2θ with a step size of 0.05°. The obtained diffractogram was compared with reference patterns from the Inorganic Crystal Structure Database (ICSD).

The surface chemistry of the MIONs was analyzed by attenuated total reflectance Fourier-transform infrared spectroscopy (ATR-FTIR) using a Vertex 70 device from Bruker (Billerica, MA, USA), after drying the sample in an oven overnight.

Thermogravimetric analysis (TGA) was performed using a TGA/DSC 1 STARe system and a gas controller 4C200 STARe system from Mettler Toledo (Columbus, OH, USA). 15 mg of the dried sample was placed in a ceramic crucible and measured at a heating rate of 10 °C/min under, an oxygen atmosphere.

The magnetic behavior and saturation magnetization of MIONs was examined via a superconducting quantum interference device (SQUID) using the MPMS-5S instrument from Quantum Design. The samples were dried analogously to the FTIR preparation. 

The optical properties of the QDs and PEGNIO/QDs/MIONs were analyzed by UV/Vis spectrometry, measuring in a 1.0 cm quartz glass cuvette using a UV-3100PC UV-Vis spectrophotometer from VWR (Radnor, PA, USA). 

The iron concentration in the niosomal samples was determined via a spectrophotometric technique. Standard solutions of Fe^3+^ containing 0.1–10 mM Fe were prepared. 10 µL of samples or standards were mixed with 500 µL of hydrochloric acid and hydrogen peroxide solution (10 mL of 6 M HCl and 10 µL of 30% H_2_O_2_). Samples were incubated with the solution for 30 min at room temperature and the optical density of the solution was measured at 410 nm using the UV-3100PC UV-Vis spectrophotometer. The iron concentrations of the samples were calculated from a calibration curve obtained from the standards. 

The colloidal stability of the niosomal formulations was tested via DLS analysis. All samples were stored at 4 °C in the dark. The particle size and PDI values were measured repeatedly over the period of one month. Moreover, the particle size of PEGNIO/QDs/MIONs/Tf was measured in cell culture media and PBS after incubation at 37 °C for 24 h.

### 3.7. Cellular Uptake and Internalization

U87 cell lines were provided from the German Collection of Microorganisms and Cell Cultures (DSMZ). U87 cells were grown in DMEM containing 10% fetal calf serum (FCS) (Biochrom GmbH, Berlin, Germany) and 1.0% penicillin/streptomycin (P/S, Sigma-Aldrich). Cells were cultivated and incubated with samples at 37 °C in a humidified environment with 5.0% CO_2_.

The uptake of the nanoparticle-loaded niosomal formulations by U87 cells was analyzed by flow cytometry. The cells (5·10^5^) were treated with niosome samples for 2 h, washed twice with PBS, and then analyzed in a BD Accuri C6 cytometer (BD Biosciences, San Jose, CA, USA) using the FL-2 channel with the excitation wavelength at 488 nm. As a control, untreated U87 cells were used. At least 20,000 gated events were observed in total, and living cells were gated in a dot plot of forward versus side scatter signals. The dot plot and histogram data were analyzed by Flowing Software 2. 

Cellular internalization of the samples was determined via fluorescence microscopy using a Cytation 5 imaging reader (BioTek Instruments Inc., Winooski, VT, USA). U87 cells were cultivated for 2 days on 96 well plates in 200 µL of the medium. The samples (PEGNIO/QDs/MIONs and PEGNIO/QDs/MIONs/Tf) were diluted with cell culture medium and then added to the cells. To evaluate the effect of an external magnetic field on cellular uptake, a neodymium magnet with a magnetic field of 1.3 T was placed under the cell-growing plate for 15 min. Afterwards, the cells were incubated for 4 h at 37 °C without an external magnetic field and washed once in PBS. The fluorescence images were evaluated by calculating the enhancement factor using the ImageJ software according to Balasubramanian et al. [56].

### 3.8. Cytotoxicity

The metabolic activity of viable cells, in terms of their reduction capacity of resazurin, was measured via CTB assay (CellTiter-Blue^®^ cell viability assay, Promega Corp., Madison, USA). The cytotoxic effects of niosomal formulations were tested on U87 cells. Cells (8·10^3^) were seeded out in 96-well tissue plates (Sarstedt, Nümbrecht, Germany) in a volume of 200 µL and cultivated for three days. After this cultivation time, the cells were washed once with PBS, treated with PEGNIO/QDs/MIONs and were exposed to an external magnetic field for 15 min by placing a neodymium magnet with a magnetic field of 1.3 T under the well plate. After further incubation for 24 h, the cells were washed twice with PBS. Then, 100 µL of CTB reagent (diluted 1: 6 with supplement-free DMEM medium) was added to each well and incubated for 1 h (37 °C, 5% CO_2_). The resulting fluorescence intensities (544Ex/590Em) were recorded with a fluorescence spectrometer (Fluoroskan Ascent, Thermo Fischer Scientific Inc., Waltham, MA, USA). Furthermore, IC_50_ values (the concentration required for 50% inhibition of cell viability) were calculated for U87 cells using the growth sigmoidal/dose-response function of Origin software. The equivalent concentration of Fe was used in the niosomal formulations.

### 3.9. MR Imaging

For MRI, 8,000 U87 cells were seeded and incubated for two days in a 6 well-plate (Sarstedt, Nümbrecht, Germany). After that, a PEGNIO/QDs/MIONs/Tf dispersion sample was applied to the cells for 2 h. Then, the medium was removed from the wells and the cells were washed with PBS twice. Next, pure medium was added into the wells, and MR imaging was carried out using the T2-weighted state of the SIGNA Explorer (1.5 T, 60 cm) by GE Healthcare (Chicago, IL, USA). Then, the amount of contrast was measured with the Image J software by generating an MRI histogram. The negative control was prepared with U87 cells using only the medium under the same conditions.

### 3.10. Statistical Analysis

The statistical evaluation of the results (cytotoxicity, fluorescence microscopy, and MRI studies) was investigated by performing the Student’s *t*-test. The difference between two groups was considered significant when the *p*-value was less than 0.05.

## 4. Conclusions

In the described approach, InP/ZnS quantum dots and CA-MIONs were integrated into shell and core, respectively, of a niosomal structure. The resulting niosomes exhibited both fluorescent and magnetic properties with low cytotoxicities. The incorporation of CA-MIONs allowed MR imaging and, simultaneously, magnetic targeting of a whole imaging system. In a further step, Tf was covalently bound to the surface of niosomes to target glioma cells overexpressing Tf receptors. For the evaluation of the specific binding efficiency and cellular localization, niosomal formulations were applied to glioma cells. Our results show that PEGNIO/QDs/MIONs/Tf bind to Tf-positive glioma cells more effectively than PEGNIO/QDs/MIONs. In vitro imaging studies revealed that PEGNIO/QDs/MIONs/Tf produce an obvious negative-contrast enhancement effect on glioma cells by magnetic resonance imaging; moreover, they present improved fluorescence intensity under fluorescence microscopy, especially with external magnet treatment. The results thus indicate the high potential of multifunctional niosomes for cell-specific dual targeting and dual imaging of glioma. To validate the in vitro results as well as investigate potential side effects, in vivo experiments with a special focus on the safe and selective delivery of the diagnostic nanocarriers will be necessary. In essence, the development of the targeted niosomal formulation reported here may provide a platform for a new generation of targeting and labeling systems.

## Data Availability

Data are available upon request.

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
