# Peer review of "Transferrin-Decorated Niosomes with Integrated InP/ZnS Quantum Dots and Magnetic Iron Oxide Nanoparticles: Dual Targeting and Imaging of Glioma"

_ijms, 2021, doi:10.3390/ijms22094556_

Round 1
Reviewer 1 Report
The paper is well written the characterization techniques were correctly chosen and the results are interesting and promising.
There are some major and minor, observations the authors might address before publication:
1.Could the authors estimate the incorporation efficiency of MIONs in niosomes?
2.the authors should consider increasing the quality of their figures
3. the authors should provide more technical details in material and methods section.
4.To confirm Tf/mediated endocytosis they should use another cell line that does not express Tf receptor or use an antibody that blocks Tf receptor (as performed in salzano et al. in this paper LN229 express higher tf receptor levels than U87)
Author Response
Reviewer 1
The paper is well written the characterization techniques were correctly chosen and the results are interesting and promising.
There are some major and minor, observations the authors might address before publication:
Comment 1: Could the authors estimate the incorporation efficiency of MIONs in niosomes?
Reply: We would like to thank the reviewer for the encouraging feedback. In a previous study on an analogous system, we determined the incorporation efficiency of MIONs into niosomes by repeatedly synthesizing MION-loaded niosomes (n = 40) and evaluating the encapsulation efficiency via photometric measurements. We thereby determined an encapsulation efficiency of MIONs into niosomes of approx. 30 %. We pointed this out in lines 319-320 in chapter 3.1: “In a recent study, we showed that the encapsulation of MIONs into the niosomal structure was achieved with an entrapment efficiency of approx. 30 %, which results in the formation of superparamagnetic niosomes with reduced saturation magnetization but still an efficient magnetic targeting capability [47].”
Comment 2: the authors should consider increasing the quality of their figures.
Reply: According to the reviewer’s suggestion, we have increased the resolution and quality of all the figures in our manuscript. Please note that the images might be displayed in lower resolution in the pdf version of the document for review.
Comment 3: the authors should provide more technical details in material and methods section.
Reply: As requested, we have revised the “Materials and Methods” section and included further technical details.
Comment 4: To confirm Tf/mediated endocytosis they should use another cell line that does not express Tf receptor or use an antibody that blocks Tf receptor (as performed in salzano et al. in this paper LN229 express higher tf receptor levels than U87).
Reply: In the internalization studies, we unveiled that the cellular uptake kinetics of our hybrid nanoparticles can be highly enhanced by using the double targeting modality. The experimental setup for our studies was based on several previously published results, such as the proof of the Tf/mediated endocytosis of Tf-functionalized niosomes by glioma cells [37, 48-51]. Since those studies already verified the Tf-receptor based enhanced internalization, we focused on the successful application and validation of the dual targeting modality. We have addressed this in our manuscript in lines 393-396: “Tf has been shown before to be a promising targeting ligand with high specificity and affinity for glioma [37, 48-51]. Since those studies already verified the Tf-receptor based enhanced internalization, no Tf-negative control cells were used for the present internalization studies.”
In summary, we have revised our manuscript taking into account the valuable suggestions by the reviewers. It is the authors’ honest opinion that the presented data and results are highly interesting for the readers of MDPI International Journal of Molecular Sciences by their completeness and careful analysis, showing that the present multifunctional nanoparticles can be a valuable source for the design, preparation and application of multimodal imaging platforms targeting various cancerous tissues.
Reviewer 2 Report
Overall revision is reasonable.
Author Response
Reviewer 2
Overall revision is reasonable.
Reply: We thank the reviewer for his/her positive feedback.
Reviewer 3 Report
As the authors addressed properly all the issues raised in the previous review I recommend the publication of the article.
There is a small error in the legend of figure 1 C, where the fluorescence and absorption spectra of InP/ZnS QDs are presented and I believe the fluorescence spectrum is the red line and the extinction one is the blue line.
Author Response
Reviewer 3
As the authors addressed properly all the issues raised in the previous review I recommend the publication of the article.
There is a small error in the legend of figure 1 C, where the fluorescence and absorption spectra of InP/ZnS QDs are presented and I believe the fluorescence spectrum is the red line and the extinction one is the blue line.
Reply: We thank the reviewer for his/her positive feedback and for pointing out this mistake. Indeed, the allocation of the extinction and fluorescence spectra in the legend of figure 1 C was wrong, which we have corrected accordingly.
Round 2
Reviewer 1 Report
As the authors addressed the issues raised in
the previous review unfortunately they should
perform experiments at least in another cell line
Author Response
Reviewer 1
As the authors addressed the issues raised in the previous review unfortunately they should perform experiments at least in another cell line.
Reply: We thank the reviewer for his/her further assessment of our manuscript. We agree that the use of Tf-receptor negative cells to prove a lower cellular uptake of our Tf-functionalized nanoparticle system would be unequivocal evidence for a Tf-based endocytosis in glioma cells. Due to the encapsulated state of the QDs and MIONs and since tests of the entire PEGNIO/QDs/MIONs/Tf system require a complex and high effort fabrication, we performed flow-cytometric measurements using PEGNIO and PEGNIO/Tf with a Tf-negative cell line (A549, which expresses hardly any Tf-receptors [54]), which clearly showed Tf-functionalization-independent cellular uptake (we have added the results to the Supporting Information section, Fig. S7). Since the increased uptake of Tf-functionalized PEGNIO formulations by Tf-receptor expressing U87 cells was proven in our manuscript, and QDs and MIONs will not affect internalization as they are encapsulated within PEGNIO, we infer an unambiguous Tf-mediated endocytosis of PEGNIO/QDs/MIONs/Tf. We have addressed this in the manuscript in lines 430-436 as follows: “We verified this Tf-receptor-mediated endocytosis of Tf-functionalized PEGNIO formulations by Tf-receptor-expressing U87 glioma cells via performing flow-cytometric cellular uptake measurements of fluorescence-labeled PEGNIO/Tf in Tf-receptor negative cells (A549, which expresses hardly any Tf-receptors [54]) (Fig. S7). Since in this case the internalization of PEGNIO and PEGNIO/Tf was shown to be almost uniform and therefore independent of Tf-functionalization, the presented enhanced internalization of Tf-labeled PEGNIO/QDs/MIONs by U87 cells can be attributed to a Tf-based endocytosis.”